# Geographical Variation of Honey Bee (*Apis mellifera* L. 1758) Populations in South-Eastern Morocco: A Geometric Morphometric Analysis

**DOI:** 10.3390/insects13030288

**Published:** 2022-03-15

**Authors:** Abdessamad Aglagane, Adam Tofilski, Omar Er-Rguibi, El-Mustapha Laghzaoui, Latifa Kimdil, El Hassan El Mouden, Stefan Fuchs, Andrzej Oleksa, Ahmed Aamiri, Mohamed Aourir

**Affiliations:** 1Laboratory of Biodiversity and Ecosystem Functioning, Faculty of Science, Ibn Zohr University, Agadir 80000, Morocco; aamiriah@gmail.com (A.A.); maourir@gmail.com (M.A.); 2Department of Zoology and Animal Welfare, University of Agriculture in Krakow, 31-425 Krakow, Poland; rotofils@cyf-kr.edu.pl; 3Laboratory of Water, Biodiversity and Climate Change, Faculty of Science, Semlalia, Cadi Ayyad University, Marrakech 40000, Morocco; omar.er.rguibi@gmail.com (O.E.-R.); laghzaoui.el@gmail.com (E.-M.L.); kimdil.latifa@gmail.com (L.K.); elmouden@uca.ac.ma (E.H.E.M.); 4Institut für Bienenkunde, Polytechnische Gesellschaft, Faculty of Biological Sciences, Goethe-Universitaet Frankfurt am Main, Karl-von-Frisch-Weg 2, 61440 Oberursel, Germany; s.fuchs@bio.uni-frankfurt.de; 5Department of Genetics, Faculty of Biological Sciences, Kazimierz Wielki University, Powstańców Wielkopolskich 10, 85-090 Bydgoszcz, Poland; olek@ukw.edu.pl

**Keywords:** geometric morphometrics, *Apis mellifera sahariensis*, Saharan bee, hybridization, introgression

## Abstract

**Simple Summary:**

Western honey bees are of high importance to human food security as they provide valuable contributions through pollination. Unfortunately, high levels of honey bee colony losses have been registered around the world recently. One of the major reasons for these losses is the hybridization with non-native subspecies which leads to the loss of adaptation to local climatic conditions. In fact, it is proven that honey bee subspecies that are native to a certain area subsist better than imported ones. In this study, we investigate the conservation status and the geographic variation of four populations of *Apis mellifera sahariensis* in south-eastern Morocco using the geometric morphometric approach. The results obtained have indicated that our samples were significantly different from the two subspecies used as reference (*Apis mellifera sahariensis*, *Apis mellifera intermissa*) which could be explained by a hybridization phenomenon occurring in the study area. The four populations studied were significantly different in terms of wing shape. These differences were mainly due to the fragmented distribution of the study area. Results of this study can be used in the planning of future strategies for the conservation of the Saharan honey bee in Morocco.

**Abstract:**

In Morocco, there are two well-recognised honey bee (*Apis mellifera* L.) subspecies: *A. m. intermissa* in the north and *A. m. sahariensis* in the south-east. The latter subspecies is found in the arid and semiarid climates of the Sahara Desert. In this study, we used honey bees from four areas of south-eastern Morocco which are, to some degree, isolated by arid zones. We analysed the shape and size of the forewings, using the method of geometric morphometrics. The bees from the four areas of south-eastern Morocco differed significantly in terms of wing shape. Moreover, bees from traditional hives were smaller than those from modern hives. The bees from south-eastern Morocco were clearly different from the reference samples obtained from the Morphometric Bee Data Bank in Oberursel, Germany, representing most of the global variation in honey bees. Surprisingly, the bees were also different from *A. m. sahariensis*, which should occur in the study area, according to earlier studies. This difference could have been caused by introgression with non-native subspecies imported by beekeepers. The distinct honey bees from south-eastern Morocco deserve to be protected. We provide a method for identifying them, which can help protect them.

## 1. Introduction

The subspecies of the western honey bee (*Apis mellifera* L.) are naturally distributed throughout Africa, Europe and western Asia [1]. Based on traditional morphometric tools, Ruttner [1] verified 24 subspecies, which he grouped into four lineages: the African lineage (A), the western Mediterranean lineage (M), the northern Mediterranean lineage (C) and the eastern lineage (O). Today, we recognize approximately 31 subspecies, 11 of which are located only in Africa [2]. According to Ruttner [1], there are two clearly distinguishable subspecies in Morocco: *Apis mellifera intermissa* (Maa, 1953) and *Apis mellifera sahariensis* (Baldensperger, 1932), while the status of a third subspecies–*Apis mellifera major* (Ruttner, 1976)–is unclear [1,3]. These subspecies were considered to belong to the western Mediterranean lineage [1]. However, mitochondrial DNA markers later confirmed that they belong to the African lineage [4].

*A. m. sahariensis*, also called the Saharan honey bee, was discovered by Philipp Baldensperger in south-eastern Morocco and western Algeria in the early 19th century [1]. Those bees differed from others in the surrounding areas with their yellow body colour and less defensive behaviour. This subspecies is endemic to the oases of south-eastern Morocco, from Ouarzazate to Figuig and to western Algeria, from Ain Sefra to Jebel Amour [1]. The first morphometric analysis of Saharan bees performed by DuPraw [5,6] showed that in terms of wing venation they are similar to *A. m. intermissa*; he even suggested that *A. m. sahariensis* and *A. m. intermissa* are not separate subspecies but colour forms [5]. Later comparisons, however, confirmed that the two subspecies could be reliably discriminated by using a large number of variables, including body colour and size [1]. When the shape of wing venation was analysed by geometric morphometrics, significant differences were found between *A. m. sahariensis* and other subspecies. However, a similarity to *A. m. intermissa* is also clearly visible and in some studies, a relatively large overlap was found between the two subspecies [7,8,9].

The Saharan honey bee was recommended for apiculture due to its less defensive behaviour, high adaptation to difficult climatic conditions and productivity [1,10]. This subspecies had been introduced to Europe by the beginning of the 20th century, though proved to be less adapted to the colder climate [1]. Later, *A. m. sahariensis* was hybridised with other subspecies by honey bee queen breeders and the hybrids were distributed by beekeepers as the ‘Buckfast’ breeding line [11].

Unfortunately, during the 1980s, populations of these subspecies faced a serious drop due to anti-locust treatment, drought and the introduction of the Varroa mite [12]. This situation forced beekeepers to buy colonies from other parts of the country in order to restore their colony stocks. The imported colonies were mainly of the subspecies *A. m. intermissa*. The massive introduction of this subspecies into the natural range of the Saharan honey bee and the practice of transhumance are two factors that may have altered the gene pool and structure of local honey bee populations, leading to hybridisation between differentiated subspecies [12,13]. To date, the current status and the geographic variation of the *A. m. sahariensis* populations in south-eastern Morocco have not been studied in-depth. Although *A. m. sahariensis* and *A. m. intermissa* are two honey bee subspecies that show deep morphometric and behavioural variance, numerous studies have detected evidence of introgression and hybridisation between them [12,14,15,16,17].

Several studies have been conducted to characterise and differentiate honey bee subspecies around the world using various approaches, including classic and geometric morphometrics as well as genetic tools [7,12,13,18,19,20]. Using classical morphometry analysis, Cornuet et al. [14] were able to prove the presence of three distinct honey bee subpopulations in Morocco. However, mtDNA testing did not clearly differentiate these three groups [4]. Likewise, a geometric morphometric analysis was conducted to differentiate *A. m. sahariensis* from its immediate neighbour, *A. m. intermissa* from Algeria and from the South African subspecies *A. m. capensis* [8].

The identification of honey bee subspecies using geometric morphometrics has proven to be a useful and more powerful method than classical morphometry [21]. Through the analysis of wing shape and size variance, several studies have been able to investigate the morphometric diversity of local subspecies and evaluate their current status [8,22,23]. Contrary to traditional morphometry, which uses distances, angles and ratios as the discriminating criteria, the geometric morphometrics approach uses a set of landmark coordinates to evaluate the differences in wing shape and size between given groups [24,25].

This study aims to investigate the conservation status and geographic variation of *A. m. sahariensis* in south-eastern Morocco. To that aim, we used the geometric morphometric approach to (1) allocate honey bee samples from south-eastern Morocco to one of the two reference subspecies (*A. m. sahariensis* and *A. m. intermissa*) and (2) evaluate the morphometric variability and the wing shape and size patterns of four populations of bees separated geographically within the natural range of the Saharan honey bee. The knowledge gained from this analysis might prove to be of high importance regarding future conservation strategies for the Saharan honey bee.

## 2. Materials and Methods

### 2.1. Sampling

Samples of worker honey bees were taken from 110 colonies between 2019 and 2020. The study area covers the south-eastern slope of the High Atlas Mountains, which is within the natural range of *A. m. sahariensis* (Appendix A and Figure 1). These samples represent four areas and 35 localities of the Darâa-Tafilalet region: (1) Zagora (26 colonies), (2) Ouarzazate (31 colonies), (3) Tinghir (33 colonies) and (4) Errachidia (20 colonies). Those areas correspond to administrative provinces with the same name, except three colonies which are in administrative Errachidia province but were included in the Tinghir area because of their proximity to this group. The choice to treat the four areas as separate groups was based on the fact that they are oases separated by relatively large arid zones, which can be an important barrier that limits gene flow between these subpopulations. Particularly well isolated are Zagora and Errachidia, which are at lower altitudes (mean about 1000 MASL) and sparsely populated by people. Those two areas are arider and, in most part, inhabitable for both humans and bees. On the other hand, Ouarzazate and Tinghir represent the western and eastern parts, respectively, of more densely populated areas which are at higher altitudes (mean about 1500 MASL). The minimum distance between locations from two neighbouring areas (point Iminoulaoune/Ouarzazate and Ait Toumort/Tinghir) is 31.4 km. This distance should be considered as relatively large because most of the area between those two locations is very arid and not suitable for bees. In two areas (Errachidia and Zagora) there were only modern hives, whereas the other two areas (Ouarzazate and Tinghir) contained both modern and traditional hives. The traditional hives were represented by eight out of 31 hives in Ouarzazate and 11 out of 33 in Tinghir. The traditional hives are typical for North Africa [26] and consist of horizontal cavities in house or garden walls made of clay. On the other hand, all of the modern hives in the study were of the Langstroth type. 

In every colony, at least 20 worker bees were collected from the nest and were immediately transferred into 96% ethanol until analysis. As reference material, we used forewing images of *A. m. intermissa* (19 colonies) and *A. m. sahariensis* (four colonies) obtained from the Morphometric Bee Data Bank in Oberursel, Germany.

### 2.2. Geometric Morphometric Analysis

The left forewings of 10 worker honey bees from each colony were detached at their base, mounted between two microscope slides and photographed using a Motic stereomicroscope equipped with a digital camera (DM-143-FBLED-A5). A total of 1100 forewing pictures were successfully obtained. The images had 2048 × 1536 pixels and a resolution of 227,863 pixels per metre. Nineteen landmark coordinates were digitised on each wing using the software IdentiFly and following the same labels and order as published in Nawrocka et al. [20]. The Cartesian coordinates of the landmarks were aligned and averaged within the colonies in the software MorphoJ 1.07 [27], using generalised Procrustes analysis [28]. The analysis allowed us to extract and visualise the variation in the wings’ shape after being scaled, translated and rotated against the consensus configuration [29]. As a measure of wing size, we used the logarithm-transformed centroid size, which is the square root of the sum of squared distances between the centre of the forewing and each of the 19 landmarks [29].

### 2.3. Statistical Analysis

The aligned landmark coordinates were analysed with principal component analysis using MorphoJ 1.07 [27] to obtain 34 principal components. The principal components were further analysed with canonical variate analysis (CVA) using PAST version 4.05 [30] to visualise the differences between the groups and to classify colonies into groups using jack-knife cross-validation. Statistically significant differences in wing shape were evaluated by the 34 principal components and were analysed using multivariate analysis of variance (MANOVA). Relationships between wing shape and latitude, longitude and altitude were analysed using multivariate regression. Relationships between two univariate variables (for example, wing size and latitude) were analysed with Pearson’s correlation. Differences in wing size were based on logarithm-transformed centroid size and were analysed by analysis of variance (ANOVA). Standard wing morphometry was also used to compare our dataset with earlier studies. Using trigonometry, we converted the coordinates of the landmarks to 16 angles (A1, A4, B4, D7, E9, G7, G18, H12, J10, J16, K19, L13, M17, N23, O26 and Q21), two vein lengths (LG and AO) and the cubital index (CuI) (for details, see DuPraw [5] and Puškadija et al. [23]).

The honey bees from south-eastern Morocco used in this study were identified using IdentiFly software [20]. The identification data were saved in a file called ‘apis-mellifera-SE_Morocco-classification.dw.xml’. The software, together with the identification file, can be downloaded from http://drawwing.org/identifly (accessed on 31 January 2022).

## 3. Results

### 3.1. Wing Shape

Wing shape (represented by the 34 principal component scores) differed significantly between areas and reference samples (MANOVA: Wilks’ lambda = 0.009; *p* < 0.0001). The graph of the first two principal components revealed that in terms of wing shape, bees from south-eastern Morocco differed to some degree from both *A. m. sahariensis* and *A. m. intermissa*, but there was some overlap between the groups (Figure 2A). The differences between the reference samples and the bees from south-eastern Morocco were even clearer on the CVA graph (Figure 2B). In pairwise comparisons, all groups significantly differed from each other (Table 1). The differences in wing shape between the bees from south-eastern Morocco and the reference samples of *A. m. sahariensis* and *A. m. intermissa* were present in all parts of the wing (Figure 3).

When wing shape was compared between traditional and modern hives in two areas, significant differences were found between Tinghir and Ouarzazate (two-factor MANOVA, area factor: Wilks’ lambda = 0.192; *p* = 0.001); no significant difference was found between traditional and modern hives (two-factor MANOVA, hive type factor: Wilks’ lambda = 0.438; *p* = 0.485). The interaction between the two factors was not statistically significant (two-factor MANOVA, area * hive type interaction: Wilks’ lambda = 0.423; *p* = 0.420). There was a significant relationship between wing shape and latitude (multivariate regression: Wilks’ lambda = 0.527, *p* = 0.009), longitude (multivariate regression: Wilks’ lambda = 0.414, *p* < 0.0001) and altitude (multivariate regression: Wilks’ lambda = 0.462, *p* = 0.0004).

According to IdentiFly, 109 out of 110 (99.09%) colonies from south-eastern Morocco were classified as belonging to the African lineage (A). One colony (0.91%), from Zagora, was classified as belonging to the Oriental lineage (O). In order to detect non-native colonies, we compared them with reference samples from the Morphometric Bee Data Bank. In the comparison we used four main lineages: C (*n* = 37), M (*n* = 16), O (*n* = 49) and A (*n* = 100). Linear Discriminant Analysis with cross-validation revealed that all colonies but one (99.09%) were correctly classified as south-eastern Morocco (Figure 4); only one colony (0.91%), from Zagora, was classified as lineage A.

### 3.2. Wing Size

Wing size (represented by a logarithm of the centroid size) differed between areas and reference samples (ANOVA: F (5127) = 11.07; *p* < 0.0001; Figure 5). In posthoc pairwise comparisons (Tukey HSD test), *A. m. intermissa* differed from all four areas of south-eastern Morocco. On the other hand, *A. m. sahariensis* was only significantly different from the samples from Zagora. No significant difference was found between the four areas except for Zagora, where wings were smaller than in Errachidia and Tinghir. When wing size was compared between traditional and modern hives in two areas, it was found that the bees from Tinghir had larger wings than those in Ouarzazate (two-factor ANOVA, area factor: F (1,60) = 8.82; *p* = 0.004) and that bees in modern hives had larger wings than those in traditional hives (two-factor ANOVA, hive type factor: F (1,60) = 5.64; *p* = 0.021). The interaction between the two factors was not statistically significant (two-factor ANOVA, area * hive type interaction: F (1,60) = 0.110; *p* = 0.741; Figure 6). When only modern hives were used in the analysis the wing size was significantly positively correlated with the latitude (Pearson correlation: r = 0.244, *p* = 0.019) and longitude (Pearson correlation: r = 0.269, *p* = 0.009), but not with altitude (Pearson correlation: r = 0.115, *p* = 0.274).

### 3.3. Standard Morphometry

In order to compare our results with earlier studies based on the cubital index, distances and angles, we calculated these measures from the coordinates of the landmarks (Table 2). The results of pairwise comparisons indicate that 11 measurements out of 19 significantly differed between the samples from south-eastern Morocco and the reference samples of *A. m. intermissa*. However, only seven of them were significantly different from the reference samples of *A. m. sahariensis* (Table 2). In six cases (A1, G18, H12, J10, N23 and AO) the mean values of the measurements from south-eastern Morocco were outside of the confidence limits reported by DuPraw [5].

## 4. Discussion

The data presented herein indicate that the honey bee samples from the south-eastern part of Morocco proved to be part of lineage A, which is in agreement with earlier findings [4,17,31]. Only one colony, from Zagora, was classified as belonging to lineage O. However, as there are no reports of queen importation specifically from lineage O and given the relatively high similarity between lineages A and O, this finding seems more likely attributable to a misclassification.

Surprisingly, the wing shape of bees from south-eastern Morocco differed markedly from the reference samples of *A. m. sahariensis*, which according to earlier research [1,5] should be found in the study area. In fact, the wing shape of our samples was more similar to *A. m. intermissa* than to *A. m. sahariensis* (Table 1). One explanation for this discrepancy can be the hybridisation of the original south-eastern Moroccan population with bees introduced from other geographic regions. It is well known that beekeepers from south-eastern Morocco buy honey bee colonies from the north of the country, where *A. m. intermissa* should occur. In fact, this result was expected as there are reports that since the 1980s, the gene pool of honey bee populations in south-eastern Morocco has been changing due to the massive importation of honey bee colonies of the subspecies *A. m. intermissa* into the natural range of the Saharan subspecies *A. m. sahariensis* [12]. Surprisingly, the present population of south-eastern Morocco was not intermediate in the CVA representation between reference samples of *A. m. sahariensis* and *A. m. intermissa*. The cluster of a ‘hybrid’ group between two honey bee subspecies is expected to present a wing venation similar to the average shape of the two subspecies of origin [32]. However, in two other studies [33,34], the hybrids presented in discriminant function graphs formed a separate cluster of points which were also not intermediate between the clusters of the original, ‘pure’ subspecies.

It is also worth mentioning that, in addition to genetics, the morphometric traits of honey bees–including wing venation–can also be affected by environmental factors [35]. In this study, there was a significant relationship between wing shape and altitude which is correlated with many environmental factors. On the other hand, there was an even stronger relationship between wing shape and longitude. Those spatial differences could be related to both evolutionary history and human intervention. A larger number of environmental variables and a larger geographic scale are required to gain better insight into this problem.

Another explanation of the observed discrepancy between bees from south-eastern Morocco and the reference samples of *A. m. sahariensis* is the small sample size used in the earlier studies, which was limited to six colonies [1] or ten workers [5]. Additionally, we only had access to four colonies of *A. m. sahariensis* from the Morphometric Bee Data Bank. The few other studies available did not provide numerical results for use in comparisons (apart from the cubital index, which is discussed later). As a further constraint, the reference sample of *A. m. sahariensis* does not cover the whole range of the study area; thus, it does not take into account the diversity of wing shapes occurring in the whole area. In fact, several studies have also pointed out that the sample size and the area covered by the reference subspecies markedly affected the results [23,36].

Concerning wing size, we also found a significant difference between the four areas of south-eastern Morocco and the reference samples of *A. m. intermissa*. The wing size of our samples was more similar to *A. m. sahariensis* and did not differ from it significantly. The wing size was affected by geographic origin as well as the type of hive. The bees in traditional hives had smaller wings than those in modern hives. This difference may be related to the use of a foundation in modern hives. The size of cells in the comb foundation is often larger than those found in naturally built combs. This influence of hive type on wing size impedes the interpretation of the results related to size, as it is unknown whether the bees from the reference samples and earlier studies were collected from modern or traditional hives.

Our results are in accordance with those published by Ruttner [1], who found that the wing size of *A. m. sahariensis* was smaller than that of *A. m. intermissa*. The same result was also mentioned by Cornuet et al. [14], indicating a gradient of mean body size that increases south to north. However, this was not confirmed in other studies related to wing size [5,6,9]; in one study the opposite relationship was even reported [8].

Regarding the cubital index, a wide range of contradictory values have been reported in the literature. Ruttner [1] reported a higher cubital index in *A. m. sahariensis* compared to *A. m. intermissa*, with values of 2.62 (*n* = 6 colonies) and 2.33 (*n* = 20 colonies), respectively. On the contrary, Cornuet et al. [14] found a smaller cubital index for *A. m. sahariensis* (2.33) than for *A. m. intermissa* (2.52). In the reference samples obtained from the Morphometric Bee Data Bank, the cubital index was higher in *A. m. sahariensis* than in *A. m. intermissa* (2.39 [*n* = 4 colonies] and 2.27 [*n* = 19], respectively). The mean value of the cubital index for south-eastern Morocco was 2.48 (*n* = 110) in our study, which is close to that of *A. m. sahariensis* in the Oberursel Bee Data Bank.

Unlike the geometric morphometric approach where the south-eastern Morocco honey bees were more similar to reference samples of *A. m. intermissa* (Table 1), the classical morphometry rather indicates higher similarity to the Saharan honey bee *A. m. sahariensis* (Table 2). Similar disagreement between the two methodologies was also reported by Miguel et al. [21]. The disagreement can be attributed to the difference in the character suites used by each method [21]. It is believed that geometric morphometrics is more suitable for wing shape analysis in comparison to angles used in classical morphometry [21,25]; however, the later methodology covers a larger number of variables related to various body parts and can detect differences not present in wings.

The evaluation of the morphometric variability of honey bee samples from the four populations of south-eastern Morocco has clearly indicated distinguishable differences between them in terms of wing shape and size. Similar results were also presented in earlier research [14]. Given the fragmented distribution of the study area (oases separated by arid zones), constituting a sufficient obstacle for natural genetic exchange, some divergence between these populations could be expected [37]. In view of the mostly similar environmental conditions, these incongruities are more likely the result of reduced gene flow between these four populations and the presence of genetic drift [38].

Locally adapted subspecies have proved to survive better in their original environment than imported ones [39]. Thus, prioritising the conservation of native honey bee subspecies presents an urgent step in preventing losses in genetic variation. To do so, the first stage in the process of conservation remains the identification of the morphometric diversity among such locally adapted subspecies [23]. In this study, we provide data based on wing measurements that can be used to assess and identify the current populations’ status, which may prove helpful for the conservation of the region’s native bees. Even if they are to some degree hybridised, they represent the natural range of the Saharan honey bee which is a distinct subspecies and deserves to be protected from further hybridisation. Moreover, the investigated bees most probably inherited some traits from *A. m. sahariensis* and survived for a few recent generations in a harsh local environment.

## 5. Conclusions

The results of this study indicate that in terms of wing shape, the reference samples of *A. m. sahariensis* and *A. m. intermissa* were statistically significantly different from the samples collected in south-eastern Morocco. This finding could indicate hybridisation occurring in the natural range of the Saharan honey bee in south-eastern Morocco. However, as we only had access to a very small sample size of the reference subspecies of *A. m. sahariensis*, further research is required to confirm the conclusion; such research should encompass the characterisation of behavioural and colony traits as well as genetic analysis. Moreover, the four populations investigated in this study exhibited significant differences in terms of wing shape. These differences were mainly due to the fragmented distribution of the study area, which constitutes a sufficient barrier to limit gene flow between the areas under study. The results of this study can be used in the planning of future strategies of conservation for the Saharan honey bee in Morocco.

## Figures and Tables

**Figure 1 insects-13-00288-f001:**
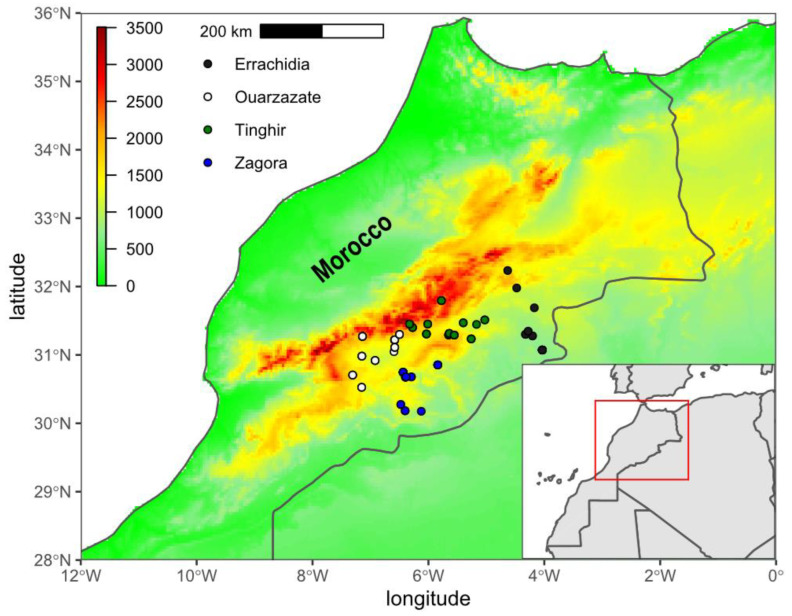
A map of Morocco with the Darâa-Tafilalet region where the sampling took place. The markers indicate the locations from which the samples were collected. Colour of the markers indi-cates four areas covered by this study.

**Figure 2 insects-13-00288-f002:**
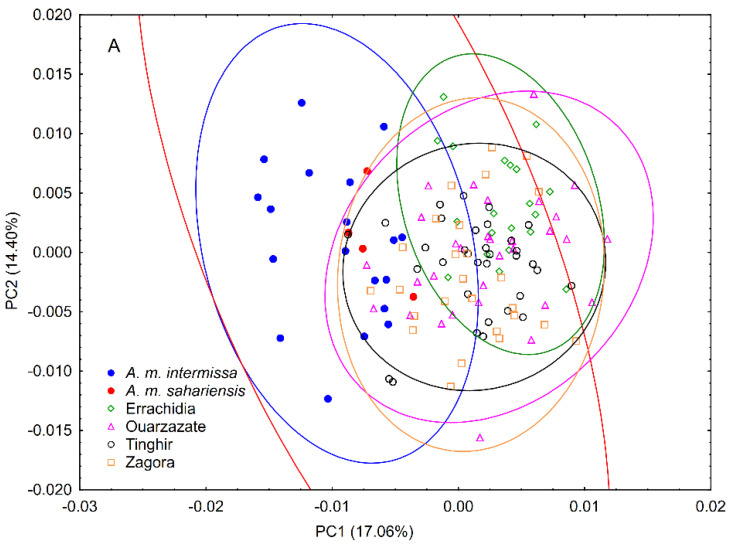
Variation of wing shape of honey bees in four areas of south-eastern Morocco in compar-ison to reference samples of *A. m. sahariensis* and *A. m. intermissa* illustrated with the first two prin-cipal components (**A**) and first two canonical variates (**B**). Each marker represents the mean scores of each colony. The ellipses represent 95% confidence intervals around the centroid of each data cluster.

**Figure 3 insects-13-00288-f003:**
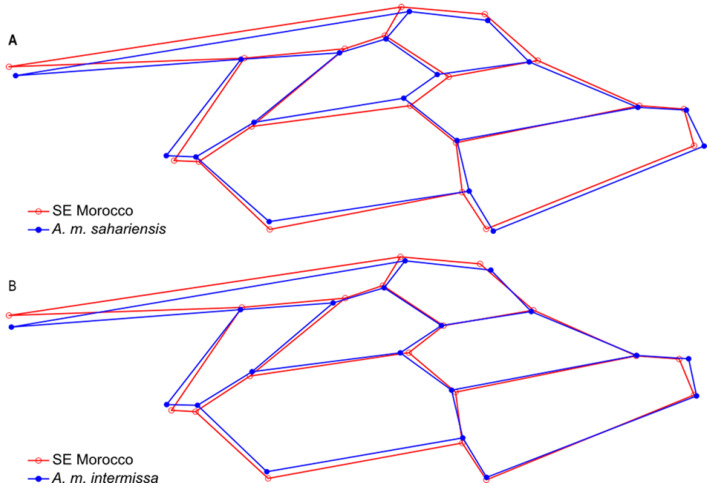
Differences between wing shape of honey bees from south-eastern Morocco and reference samples of *A. m. sahariensis* (**A**) and *A. m. intermissa* (**B**) illustrated with wireframe diagrams. The differences were increased by 3-fold for better visualization.

**Figure 4 insects-13-00288-f004:**
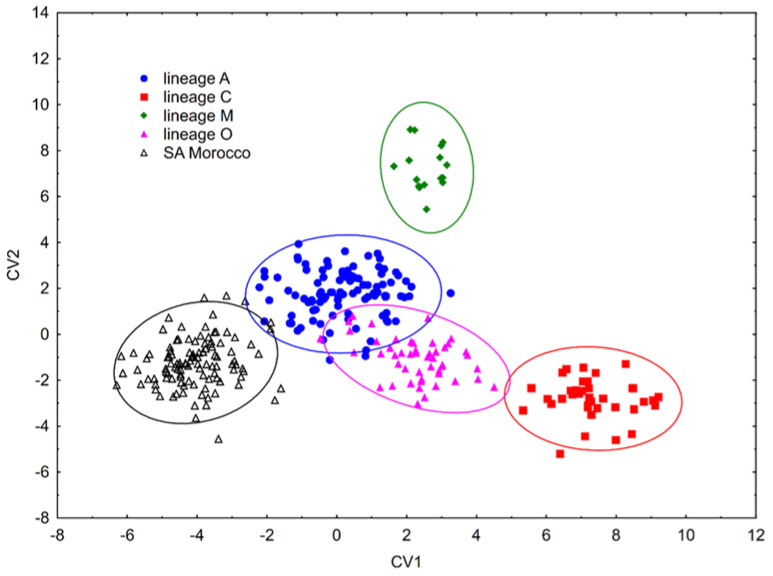
The first two canonical variates illustrating variation of wing shape of honey bees in south-eastern Morocco in comparison to reference samples of four main lineages; A, C, M, O. Each marker represents the mean scores of each colony. The ellipses represent 95% confidence intervals around the centroid of each data cluster.

**Figure 5 insects-13-00288-f005:**
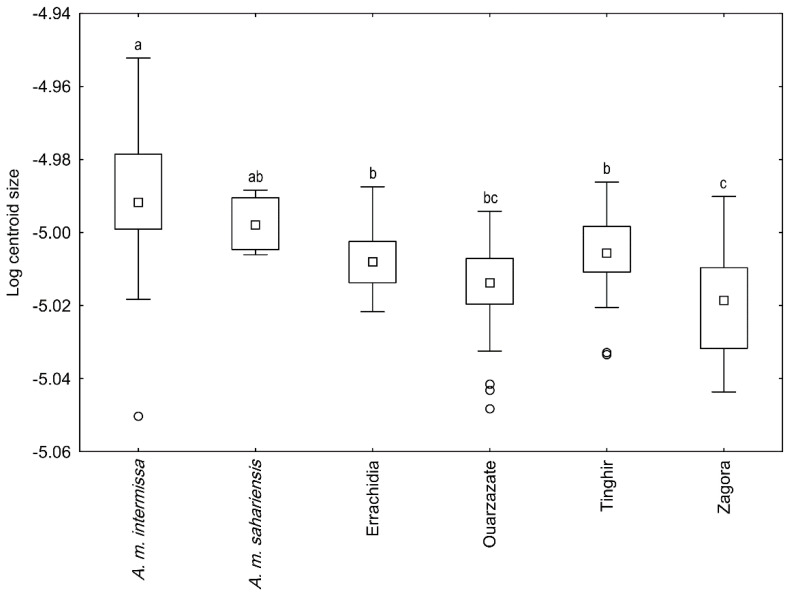
Differences in wing size between honey bees from four areas of south-eastern Morocco and reference samples of *A. m. sahariensis* and *A. m. intermissa*. Boxes indicate 25th and 75th percen-tiles; whiskers indicate the range; square markers indicate medians and circles indicate outliers. The same letters above whiskers indicate no significant difference.

**Figure 6 insects-13-00288-f006:**
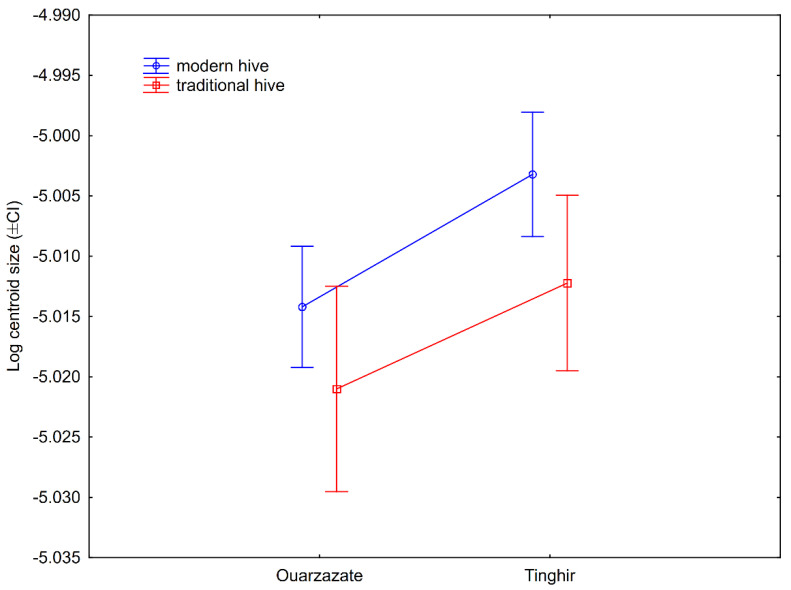
Difference between wing size (±95% confidence intervals) in modern and traditional hives in two areas of south-eastern Morocco.

**Table 1 insects-13-00288-t001:** Mahalanobis distances (lower triangle) and significance of pair-wise statistical differences (upper triangle) in wing shape of honey bees from four areas of south-eastern Morocco and reference samples of *A. m. sahariensis* and *A. m. intermissa*.

Group	Errachidia	Ouarzazate	Tinghir	Zagora	*A. m. intermissa*	*A. m. sahariensis*
Errachidia	-	<0.0001	<0.0001	<0.0001	<0.0001	<0.0001
Ouarzazate	3.4948	-	<0.0001	<0.0001	<0.0001	0.0001
Tinghir	3.2692	2.2285	-	<0.0001	<0.0001	<0.0001
Zagora	4.1742	2.7590	2.6875	-	<0.0001	<0.0001
*A. m. intermissa*	8.2451	7.2371	6.5403	6.0323	-	<0.0001
*A. m. sahariensis*	10.1513	10.0084	9.2805	8.5097	6.6533	-

**Table 2 insects-13-00288-t002:** Wing shape of honey bees from four areas of SE Morocco and reference samples of *A. m. sahariensis* and *A. m. intermissa* described by traditional morphometric indexes. For comparison we provide data for bees from Sahara published by DuPraw [5]. P1 and P2 indicates significance of differences in pairwise comparisons between samples from southeastern Morocco and *A. m. intermissa* and *A. m. sahariensis* respectively. SD–standard deviation, CI–95% confidence interval. The asterisk (*) indicate significant difference at *p* < 0.05.

Measurement	Errachidia (±SD)	Ouarzazate (±SD)	Tinghir (±SD)	Zagora (±SD)	Mean of SE Morocco (±SD)	*A. m. intermissa* (±SD)	*A. m. sahariensis* (±SD)	P1	P2	DuPraw 1965a (±CI)
CuI	2.53 ± 0.25	2.46 ± 0.29	2.58 ± 0.21	2.35 ± 0.24	2.48 ± 0.25	2.27 ± 0.18	2.39 ± 0.25	0.002 *	0.730	-
A1	26.63 ± 1.31	25.23 ± 1.59	26.02 ± 1.38	24.66 ± 1.7	25.64 ± 1.49	24.12 ± 2.3	24.53 ± 1.7	0.002 *	0.460	22.7 ± 1.4
A4	31.41 ± 1.21	32.08 ± 1.48	31.90 ± 1.08	32.03 ± 1.45	31.85 ± 1.30	31.58 ± 1.17	31.78 ± 1.82	0.610	0.980	31.1 ± 1.5
B4	106.15 ± 2.67	105.35 ± 3.49	105.36 ± 2.82	104.43 ± 2.82	105.32 ± 2.95	103.12 ± 3.62	102.47 ± 4.6	0.018 *	0.190	102 ± 4.3
D7	99.65 ± 1.33	100.84 ± 1.58	99.96 ± 1.28	100.11 ± 2.00	100.14 ± 1.55	99.99 ± 2.14	100.59 ± 1.09	0.880	0.880	-
E9	20.25 ± 0.62	20.56 ± 0.91	20.05 ± 0.64	20.35 ± 0.85	20.3 ± 0.75	19.95 ± 0.73	18.89 ± 0.34	0.160	0.001 *	19.8 ± 0.8
G7	23.61 ± 0.34	23.46 ± 0.63	23.51 ± 0.47	23.28 ± 0.71	23.46 ± 0.54	23.28 ± 0.66	22.89 ± 0.23	0.410	0.120	23.4 ± 0.8
G18	99.62 ± 1.78	99.47 ± 2.01	99.88 ± 1.44	100.52 ± 1.92	99.87 ± 1.79	99.51 ± 2.44	99.14 ± 1.81	0.720	0.730	97.5 ± 1.2
H12	18.73 ± 0.86	18.9 ± 0.97	18.66 ± 0.96	18.47 ± 1.1	18.69 ± 0.97	18.2 ± 1.55	17.51 ± 1.24	0.150	0.080	17.5 ± 0.9
J10	51.48 ± 1.78	51.09 ± 1.48	51.55 ± 2.00	51.53 ± 1.88	51.41 ± 1.78	48.7 ± 2.87	49.96 ± 1.36	0.000 *	0.319	54.1 ± 1.6
J16	89.54 ± 1.69	89.7 ± 2.36	90.3 ± 1.94	90.28 ± 2.28	89.96 ± 2.07	93.74 ± 2.00	93.75 ± 2.55	0.000 *	0.001 *	97.2 ± 3.0
K19	80.31 ± 1.36	79.73 ± 1.55	80.65 ± 1.81	78.76 ± 1.73	79.86 ± 1.61	77.79 ± 1.97	77.61 ± 2.13	0.000 *	0.039 *	-
L13	17.32 ± 0.91	17.21 ± 0.89	17.04 ± 0.6	17.44 ± 0.87	17.25 ± 0.82	15.49 ± 0.92	16 ± 0.42	0.000 *	0.010 *	-
M17	44.62 ± 1.18	45.53 ± 1.81	45.31 ± 1.33	45.07 ± 1.72	45.13 ± 1.51	44.27 ± 2.63	42.26 ± 1.02	0.090	0.003 *	46.4 ± 2.0
N23	79.92 ± 1.28	81.2 ± 1.98	81.57 ± 1.41	81.34 ± 1.48	81.01 ± 1.54	83.28 ± 2.4	84.2 ± 2.67	0.000 *	0.003 *	94 ± 1.9
O26	36.03 ± 1.77	36.48 ± 2.28	37.12 ± 2.10	36.84 ± 2.4	36.62 ± 2.14	40.12 ± 2.62	35.08 ± 2.38	0.000 *	0.340	37.4 ± 2.9
Q21	34.79 ± 0.56	34.96 ± 0.67	35.16 ± 0.78	35.49 ± 0.94	35.10 ± 0.74	34.44 ± 1.06	34.35 ± 0.65	0.003 *	0.160	35.9 ± 0.9
LG	1.99 ± 0.02	1.96 ± 0.03	1.97 ± 0.03	1.95 ± 0.03	1.97 ± 0.03	1.99 ± 0.05	1.99 ± 0.02	0.050	0.420	1.99 ± 0.03
AO	4.33 ± 0.04	4.28 ± 0.06	4.33 ± 0.05	4.28 ± 0.07	4.31 ± 0.06	4.42 ± 0.1	4.42 ± 0.05	0.000 *	0.004 *	4.44 ± 0.04

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
