# Peer review of "Geographical Variation of Honey Bee (Apis mellifera L. 1758) Populations in South-Eastern Morocco: A Geometric Morphometric Analysis"

_insects, 2022, doi:10.3390/insects13030288_

Round 1

Reviewer 1 Report

The manuscript “Geographical variation of honey bee (Apis mellifera L, 1758) populations in south-eastern Morocco: A geometric morphometric analysis” investigates if there are geographic differences in wing size and shape among four populations in south-eastern Morocco and species reference samples. This manuscript adds to our current knowledge of honey bee wing shape and potentially identifies populations to conserve. Overall, this paper is well written, and the methods are straightforward. However, the results they reported and the conclusion they draw could use more explanation. Also, it is unclear why they used political boundaries of providence to separate populations instead of some biologically relevant metric. Below are more detailed comments.

Line 29: Unclear what results the statement “the four populations studied were significantly different in terms of wing shape and size” is based on. The results show that the size and shape of the four populations overlap. The posthoc comparisons of wing size only describe differences from reference samples and not among the four populations.

Line 30: Also unclear wherein the results the differences due to fragmentation were shown. Some of the sites between these populations are closer than those within a population.

Line 121: Dividing populations up based on providences does not seem like a great delineator to use. Biological patterns are not restricted by political boundaries. As mentioned before, many of the sites between these populations are closer than those within a population. May want to consider a new way to subset these samples into different populations.

Figure 5. Added symbols or letters to denote the groups identified by the posthoc analysis.

Line 271: Unclear how genetic dominance would cause hybrids to cluster differently than ‘pure’ subspecies. Wouldn’t this cause wing shape to be similar to maternal or paternal subspecies?

Line 269: What are the conservation implication of these populations being hybrids? Would these populations still need to be conserved?

Line 273: The idea that differences may be environmentally caused needs to be expanded upon. Why is this not the most likely explanation for these populations being different from the reference subspecies? This is also the possible explanation for the following paragraph: the low number of reference samples possibly explains the differences between these populations and the reference samples.

Line 311: Unclear the analysis that shows these populations have distinguishable differences between them in terms of wing shape. As mentioned before, it is unclear what results show this as well as how the populations denoted are separate populations; the divisions between them seem arbitrary outside of them being from different providences.  

Line 235: There needs to be a better argument for why these populations need to be conserved if they are just hybrids or morphological differences are just induced by the environment.

Reviewer 2 Report

This is a well written and well presented paper, which is generally clear.  It is interesting and topical in the light of interest in preserving native honey bee species which are well adapted to their environment. The methods and procedures followed appear to be appropriate and conclusions justified.

I have noted minor points to correct, marked on the commented version of the submitted manuscript. Most of these are small points of English. I also suggest adding some comparison of the results from the geometric and traditional morphometric analyses, as to some extent these are contradictory.  A word should be deleted at line 60. Some clarification is needed at lines 113 and 141. The legends of some figures should be expanded. The method for pairwise comparisons should also be stated.

In the supplementary file, the table should be labelled as “Supplementary table 1” on each page, not  “Supplement table 1”.

Round 2

Reviewer 1 Report

The authors have made sufficient revisions for publications. The paper could be stronger with a  more rigorous explanation for how the different areas were defined, with metrics such as distance or environmental resistance. The map in figure 1 could have the land's topography to illustrate that these areas make up a distinct population. The potential environmentally induced difference in wing shape warrants more discussion. Environmental differences could be the primary driver of the differences between reference samples not from these regions and those sampled in the study.

Author Response

Dear reviewer,

Thank you very much for the valuable comments provided. We have corrected the manuscript as suggested. More explanations were added regarding the choice of the areas (please see line 133-137), and map of figure 1 was changed accordingly. In addition, the environmental conditions (latitude, longitude and altitude) were also investigated and discussed accordingly.